



**Roles of Climate Variability on the Rapid Increase of Winter Haze**
**Pollution in North China after 2010**
Yijia Zhang[1], Zhicong Yin[123]*, Huijun Wang[123]
[1]Key Laboratory of Meteorological Disaster, Ministry of Education / Joint International Research Laboratory of Climate and
Environment Change (ILCEC) / Collaborative Innovation Centre on Forecast and Evaluation of Meteorological Disasters
(CIC-FEMD), Nanjing University of Information Science & Technology, Nanjing 210044, China
[2]Southern Marine Science and Engineering Guangdong Laboratory (Zhuhai), Zhuhai, China
[3]Nansen-Zhu International Research Centre, Institute of Atmospheric Physics, Chinese Academy of Sciences, Beijing, China
*Correspondence to*: Zhicong Yin (yinzhc@163.com)
**Abstract.** North China experiences severe haze pollution in early winter, resulting in many premature deaths and considerable
economic losses. The number of haze days in early winter in North China (HD$_{NC}$) increased rapidly after 2010 but declined
slowly before 2010, reflecting a trend reversal. Global warming and emissions were two fundamental drivers of the long-term
increasing trend of haze, but no studies have focused on this trend reversal. The autumn SST in the Pacific and Atlantic,
Eurasian snow cover and central Siberian soil moisture, which exhibited completely opposite trends before and after 2010,
were proven to stimulate identical trends of meteorological conditions related to haze pollution in North China. Numerical
experiments with a fixed emission level confirmed the physical relationships between the climate drivers and HD$_{NC}$ during
both decreasing and increasing periods. These external drivers induced a larger decreasing trend of HD$_{NC}$ than the observations,
and combined with the persistently increasing trend of anthropogenic emissions, resulted in a realistic slowly decreasing trend.
However, after 2010, the increasing trends driven by these climate divers and human emissions jointly led to a rapid increase
in HD$_{NC}$.
**Keywords**: haze, PM$_{2.5}$, trend reversal, anthropogenic emission, climate variability
**1 Introduction**
Haze pollution, characterized by low visibility and a high concentration of fine particulate matter (PM$_{2.5}$), has become a
serious environmental and social problem in China, as haze dramatically endangers human health, ecological sustainability
and economic development (Ding and Liu, 2014; Wang and Chen, 2016). Exposure to PM$_{2.5}$ was estimated to cause 4.2 million
premature deaths worldwide in 2015 (Cohen et al., 2017), and in China, PM$_{2.5}$ caused up to 0.96 million premature mortalities
in 2017 (Lu et al., 2019). Air pollution accounts for a loss of 1.2–3.8% of the gross national product (GNP) annually (Zhang
and Crooks, 2012). The most polluted areas in China are North China (NC; 34–42°N, 114–120°E), Fenwei Plain, Sichuan
Basin and Yangtze River Delta; among them, NC is the most polluted (Yin et al., 2015). Meteorological conditions
characterized by low surface wind speeds and a shallow boundary layer result in stagnant air, which limits the horizontal and





vertical dispersion of particles and induces the accumulation of pollutants (Niu et al., 2010; Wu et al., 2017; Shi et al., 2019).
High relative humidity favors the hygroscopic growth of pollutants (Ding and Liu, 2014; Yin et al., 2015), and anomalous
ascending motions weaken the downward invasion of cold and clear air from high altitudes (Zhong et al., 2019). The
forecasting of meteorological conditions is more accurate on the synoptic scale, but the predictions of interannual variations
are not good enough. Thus, the prediction of haze is a considerable challenge.

36        Previous studies proved that the interannual to decadal variations in winter haze have strong responses to external forcing

factors, such as the sea surface temperature (SST) in the Pacific and Atlantic, snow cover and soil moisture (Xiao et al., 2015;
Yin and Wang, 2016a, b; Zou et al., 2017). Anomalies of these factors exerted their impacts to modulate local dispersion
conditions by atmospheric teleconnections and greatly intensified haze pollution in NC. The eastern Atlantic/western Russia
(EA/WR), western Pacific (WP) and Eurasia (EU) patterns served as effective atmospheric bridges linking distant and
preceding external factors to the anomalous anticyclonic circulations over Northeast Asia (Yin and Wang, 2017; Yin et al.,
2017). With enhanced anticyclonic anomalies, the haze pollution in NC was significantly aggravated by poor ventilation
conditions and high moisture.

44        The long-term trend of haze pollution has always been attributed to increasing human activities directly related to aerosol

emissions (Yang et al., 2016; Li et al., 2018). It is true that emissions are important in the formation of haze, but their role
varies from region to region (Mao et al., 2019). The trend of haze days in Yangtze River Delta and Pearl River Delta was
closely related to the trend of particle emissions (Fig. S1b, c), while a weak correlation existed in Fenwei Plain (Fig. S1d). A
surprising phenomenon can be seen in NC: the number of winter haze days and particle emissions showed similar trends before
early 1990s, but afterward, their close relationship disappeared (Fig. S1a). Many recent studies also showed that the long-term
trend in the haze problem has likely been driven by global warming (Horton et al, 2014; Cai et al., 2017). Weakening surface
winds have been reported over land in the last few decades while the global surface air temperature (SAT) has warmed
significantly (Mcvicar et al., 2012). In addition, enhanced vertical stability, which favors the accumulation of pollutants, has
been observed with global warming (Liu et al., 2013). However, none of the above studies focused on the change in the haze
trend. Over the past few decades, the global and Northern Hemispheric SAT averages generally displayed a continuous
warming trend, which was not exactly similar to the trend of haze days in NC (Fig. S2). It follows that haze pollution, especially
the change in its trend, is regulated by multiple drivers and that the long-term impacts of external climate forcings, which
efficiently modulate the interannual and decadal variations in haze, deserve further investigation.
**2 Datasets and Methods**
**2.1 Data description**





Monthly mean meteorological data from 1979 to 2018 were obtained from NCEP/NCAR reanalysis datasets (2.5°×2.5°),
including the geopotential height at 500 hPa (H500), vertical wind from the surface to 150 hPa, surface air temperatures (SAT),
wind speed, and special humidity at the surface (Kalnay et al., 1996). The boundary layer height (BLH, 1°×1°) values were
from Interim reanalysis data (ERA-Interim) obtained from the European Centre for Medium-Range Weather Forecasts
(ECMWF) (Dee et al., 2011). The number of haze days was calculated from the long-term meteorological data, mainly based
on observed visibility and relative humidity (Yin et al., 2017). The $PM_{2.5}$ concentrations from 2009 to 2016 were acquired
from the US embassy, and those from 2014 to 2018 were from China National Environmental Monitoring Centre. Monthly
total emissions of BC, $NH_3$, $NO_x$, OC, $SO_2$, $PM_{10}$ and $PM_{2.5}$ are obtained from the Peking University emission inventory. The
monthly mean extended reconstructed SST data (2°×2°) were obtained from the National Oceanic and Atmospheric
Administration (Smith et al., 2008). The monthly snow cover data were supported by the Rutgers University (Robinson et al.,
1993). And the monthly soil moisture data (0.5°×0.5°) were downloaded from NOAA's Climate Prediction Center (Huug et
al., 2003)
**2.2 Geos-Chem description and experimental design**
We used the GEOS-Chem model to simulate $PM_{2.5}$ concentrations (http://acmg.seas.harvard.edu/geos/). The GEOS-
Chem model was driven by MERRA-2 assimilated meteorological data (Gelaro et al., 2017). The nested grid over Asia (11°S–
55°N, 60–150°E) had a horizontal resolution of 0.5° latitude by 0.625° longitude and 47 vertical layers up to 0.01 hPa. The
GEOS-Chem model includes fully coupled $O_3$-NOx-hydrocarbon and aerosol chemical mechanisms with more than 80 species
and 300 reactions (Bey et al., 2001; Park et al., 2004). The $PM_{2.5}$ components simulated in GEOS-Chem include sulfate, nitrate,
ammonium, black carbon and primary organic carbon, mineral dust, secondary organic aerosols and sea salt.
In this study, we designed two kinds of experiments. One was an experiment for simulating $PM_{2.5}$, and the other was a
composite using simulated data. The simulation had changing meteorological fields in winter from 1980 to 2018 and the fixed
emissions in 2010 representing a high emission level. The emissions data in 2010 were from MIX 2010 (Li et al., 2017). The
numerical experiment was performed to examine the variation of $PM_{2.5}$ in the meteorological parameters during 1980–2018
under fixed-emission scenarios.
The composite was conducted to analyze the differences in the simulated $HD_{NC}$ according to the years selected for the
external forcing factors. Using the simulated dataset with the fixed-emission scenario was capable of eliminating the impacts
of emissions and simply considering the effect of the four external forcing factors. The four (two) years with the largest (Favor
Years) and smallest (Unfavor Years) four external forcing indices (i.e., $SST_P$, $-1×SST_A$, Snowc and $-1×$Soilw) were selected,
and the differences in the simulated $HD_{NC}$ under these four conditions in P1 (P2) were calculated. The simulated $HD_{NC}$ in
Favor Years minus the simulated $HD_{NC}$ in Unfavor Years was calculated to analyze the effect of these four forced factors.



### 2.3 Statistical methods

In this study, the statistical model of fitted $HD_{NC}$ was built based on MLR. This approach, a model-driven method, was ultimately expressed as a linear combination of $K$ predictors ($x_i$) that could generate the least error of prediction $\tilde{y}$ (Wilks, 2011). With coefficients $\beta_i$, intercept $\beta_0$, and residual $\varepsilon$, the MLR formula can be written in the following form: $\tilde{y}$ =$\beta_0+\sum \beta_i x_i+\varepsilon$.

The trends calculated in this study were obtained by linear regression after a 5-year running average. This method removed the interannual variation and more prominent trend characteristics. Moreover, the stage trends were calculated according to the inflection point, which passed the Mann-Kendall test.

### 3 Trend change of early winter haze

Throughout the winter in North China, the haze pollution in early winter is the most serious (Yin et al., 2019). The number of haze days in early winter in North China ($HD_{NC}$) reached a remarkable inflection point in 2010 (Fig. 1a), passing the Mann-Kendall Test. The trend of $HD_{NC}$ was vastly different before and after 2010: slowly decreased during 1991–2010 (P1) with a rate of 4.67 days/10 yr but rapidly increased after 2010 (P2, 2010–2018) with a rate of 25.43 days/10 yr. Recent studies generally revealed that based on observations, the number of boreal winter haze days across NC had a slightly decreasing trend after 1990 (Ding and Liu, 2014; He et al., 2019; Mao et al., 2019; Shi et al., 2019), which is consistent with the decreasing trend presented by the dataset in our research. In addition, Dang and Liao (2019) confirmed the varying trend of $HD_{NC}$ via simulations of the global 3-D chemical transport (GEOS-Chem) model; using the well-simulated frequency of serious haze days in winter, they also revealed the abovementioned changing trend of $HD_{NC}$, i.e., decreasing in the early stage and increasing in the later stage. To further determine the reliability of the post-2010 upward trend of $HD_{NC}$, we used hourly $PM_{2.5}$ concentrations observed at the US embassy in Beijing from 2009 to 2017 and those monitored by China National Environmental Monitoring Centre from 2014 to 2018 to count the number of days when the $PM_{2.5}$ concentrations were >75 µg m$^{-3}$ and >100 µg m$^{-3}$ (Fig. 1a). These statistics also reflected the rising trend after 2010, as well as the improved air quality in 2017 and a rebound in pollution in 2018. Although there was a certain gap between $HD_{NC}$ (basing on visibility and humidity) and these statistics, the two datasets revealed the same variations after 2010, and the statistics confirmed the robustness of the observed $HD_{NC}$.

The above analysis substantiated the rapid aggravation of haze pollution in early winter after 2010. With regard to the increase in air pollution, there is no doubt that anthropogenic emissions were the fundamental cause of this long-term variation. Before the mid-2000s, the particle emissions throughout NC sustained stable growth but gradually began to decline afterward, which is inconsistent with the trend of $HD_{NC}$ or even contrary in some subperiods. The previous decreasing trend of $HD_{NC}$ hid the effects of the increased pollutant emissions; thus, people ignored the pollution problem and failed to control it in time. As





a consequence, the subsequent rise in $HD_{NC}$ was extremely rapid and seriously harmed the biological environment and human
health. The stark discrepancy between the trends of pollutant emissions and $HD_{NC}$ strongly indicate that anthropogenic
emissions were not the only factor leading to a sharp deterioration in air quality after 2010 (Wei et al., 2017; Wang 2018).
Therefore, an important question must be asked: in addition to human activities, what factors caused the rapidly increasing
trend of $HD_{NC}$ after 2010?

125       As mentioned above, local meteorological factors could modulate the capacity to disperse and the formation of haze

particles, which have critical influences on the occurrence of severe haze pollution. To reveal the impacts of meteorological
conditions on the changing trend of $HD_{NC}$, the area-averaged linear trends of these meteorological factors in NC during P1 and
P2 were calculated, all of which exceeded the 95% confidence level (Fig. 2). In P1, the area-averaged linear trends of the
boundary layer height (BLH), wind speed and omega all showed significant positive trends, while specific humidity showed a
significant negative trend in NC; these conditions favored a superior air quality (Niu et al., 2010; Ding and Liu, 2014; Yin et
al., 2017; Shi et al., 2019; Zhong et al., 2019). However, the trends of these four meteorological factors completely reversed
in P2. Reductions in the BLH and wind speed, the enhancement of moisture, and an anomalous descending motion resisted
the vertical and horizontal dispersions of particles and helped more pollutants gather in relatively narrow spaces. These four
meteorological factors expressed an evident influence on the change trend of $HD_{NC}$ and showed reversed trends between P1
and P2, similar to $HD_{NC}$. Furthermore, the magnitudes of the change rates of these factors were stronger in P2 than in P1 (Fig.
2), and $HD_{NC}$ displayed this feature as well. The GEOS-Chem simulations with changing emissions and fixed meteorological
conditions failed to reproduce the change trend of haze (Dang and Liao, 2019). We designed an experiment driven by changing
meteorological conditions in winter from 1980 to 2018 and fixed emissions at the relatively high 2010 level. According to the
technical regulation on the ambient air quality index (Ministry of Ecology and Environment of the People's Republic of China,
2012), a haze day was defined as a day with daily mean $PM_{2.5}$ concentration exceeding 75 μg m$^{-3}$. The simulations of the
frequency of haze days in NC by GEOS-Chem reproduced the trend reversal of haze pollution (Fig. 1b). The simulation results
were highly correlated with $HD_{NC}$ and showed the feature that the trend in P2 was stronger than that in P1, indicating that
meteorological conditions drove the trend change of haze pollution.
**4 Climate variability drove the trend reversal**

145       According to many previous studies, the variabilities of the Pacific SST, Atlantic SST, Eurasian snow cover and Asian

soil moisture played key roles in the interannual variations in haze pollution in NC (Xiao et al., 2015; Yin and Wang, 2016a,
b; Zou et al., 2017), and the associated physical mechanisms were evidently revealed. Thus, the following question is raised
here: did these four factors drive the trend reversal of $HD_{NC}$, and if so, how?



As shown in Figure S3a, the preceding autumn SST in the Pacific, associated with the detrended $HD_{NC}$, presented a
Pacific Decadal Oscillation (PDO)-like "triple pattern" with two significant positive regions and one nonsignificant negative
region (Yin and Wang, 2016a; Zhao et al., 2016). In the following research, the SST anomalies in the two positively correlated
regions located in the Gulf of Alaska (40–60°N, 125–165°W) and the central eastern Pacific (5–25°N, 160°E–110°W) were
used to represent the effects originating from the North Pacific. The area-averaged September-November SST of these two
regions was calculated as the $SST_P$ index, and the correlation coefficients with $HD_{NC}$ were 0.59 and 0.67 before and after
removing the linear trend during 1979–2018, respectively; both correlation coefficients were above the 99% confidence level.
The responses of the atmosphere to these positive $SST_P$ anomalies were the positive phase of the EA/WR pattern and the
enhanced anomalous anticyclone center over NC (Yin et al., 2017; Fig. S4). Modulating by such large-scale atmospheric
anomalies, increased moisture, anomalous upward motion and reduced BLH and wind speed (Fig. S4) created a favorable
environment for the accumulation of fine particles (Niu et al., 2010; Ding and Liu, 2014; Shi et al., 2019; Zhong et al., 2019).
A numerical experiment based on the Community Atmosphere Model version 5 (CAM5) effectively reproduced the observed
enhanced anticyclonic anomalies over Mongolia and North China in response to positive PDO forcing, which resulted in an
increase in the number of wintertime haze days over central eastern China (Zhao et al., 2016). The trend changes of the North
Pacific SST were examined in P1 and P2. Consistent with the changing trend of $HD_{NC}$, reversed trends were also found in the
North Pacific, i.e., a significant negative trend during P1 and a positive trend during P2 over the two Pacific areas (Fig. 3a, b).
These similar trend changes suggest that the North Pacific SST might have been a major driver of the abrupt change in $HD_{NC}$.
It is clear that $SST_P$ underwent a significant trend change around 2010 (Fig. 4a). Thus, the persistent decline in $SST_P$ during P1
(at a significant rate of –0.2 °C/10 yr; Table 1) contributed to the slowly decreasing trend of $HD_{NC}$ (Fig. 4a) via the modulations
of $SST_P$ on the atmospheric circulation (Fig. S4). During P2, the larger increase in $SST_P$ at a rate of 2.0 °C/10 yr dramatically
drove the rapid increase in $HD_{NC}$.
Besides the triple pattern in the Pacific, two areas exhibiting significant negative correlations with $HD_{NC}$ were examined
in the Atlantic (Shi et al., 2015; Shi et al., 2015): one located over southern Greenland (50–68°N, 18–60°W) and another
located over the equatorial Atlantic (0–15°N, 30–60°W; Fig. S3a). The area-averaged September-November SST of the two
negatively correlated regions in Atlantic was defined as the $SST_A$ index, whose correlation coefficients with $HD_{NC}$ were –0.55
and –0.64 from 1979 to 2018 before and after detrending, respectively (above the 99% confidence level). The response of
atmospheric circulation to these negative $SST_A$ anomalies culminated in a positive EA/WR pattern, and the stimulated easterly
weakened the intensity of East Asian jet stream (EAJS) in the high troposphere (Fig. S5). Influenced by the colder $SST_A$, there
was a very obvious abnormal upward movement above the boundary layer, reducing both the BLH and the surface wind speed;
thus, pollutants were prone to gather, causing haze pollution (Niu et al., 2010; Wu et al., 2017; Shi et al., 2019). With a linear
barotropic model, Chen confirmed the important role of subtropical northeastern Atlantic SST anomalies in contributing to the



anomalous anticyclone over Northeast Asia and anomalous southerly winds over NC, which enhanced the accumulation of
pollutants (Chen et al., 2019). The spatial linear trend in the SST of both Atlantic areas changed from positive in P1 to negative
in P2, which was completely contrary to the trend of $HD_{NC}$ (Fig. 3a, b). The $SST_A$ reached a infection point in 2010 (Fig. 4b)
and contributed to the falling of $HD_{NC}$ during P1 (change rate of $SST_A$ = 0.55 °C/10 yr) and the rising of $HD_{NC}$ during P2
(change rate of $SST_A$ = –0.52 °C/10 yr).

185       The effect of Eurasian snow cover on the number of December haze days in NC intensified after the mid-1990s (Yin and

Wang, 2018). The roles of extensive boreal Eurasian snow cover were also revealed by numerical experiments via the
Community Earth System Model (CESM): positive snow cover anomalies enhanced the regional circulation mode of poor
ventilation in NC and provided conducive conditions for extreme haze (Zou et al., 2017). The correlation between the October-
November snow cover and $HD_{NC}$ was significantly positive in eastern Europe and western Siberia (46–62°N, 40–85°E, Fig.
S3b), where the spatial linear trend of snow cover was consistent with that of $HD_{NC}$. A significant negative trend in P1 and a
positive trend in P2 were discovered (Fig. 3c, d). The area-averaged October-November snow cover over eastern Europe and
western Siberia was defined as the Snowc index, and its correlation coefficients with $HD_{NC}$ were 0.43 and 0.54 from 1979 to
2018 before and after detrending, respectively (above the 99% confidence level). The features of the weakened EAJS and
significant anomalous anticyclone could be found clearly in the induced atmospheric anomalies associated with the positive
Snowc anomalies (Fig. S6). The related abnormal upward motion restricted the momentum to the surface. In addition, the
corresponding lower BLH and weaker surface wind speed also reduced the dispersion capacity, resulting in the generation of
more haze pollution (Fig. S6). The Snowc index fell slowly until 2010 (with a rate of –1.8%/10 yr) and then rose rapidly (with
a rate of 28.3%/10 yr) and experienced a large trend reversal in 2010, in accordance with the behavior of $HD_{NC}$ (Fig. 4c).
Therefore, relying on the revealed physical mechanisms, the strengthened relationship between Snowc and $HD_{NC}$ and the
tremendous increase in Snowc during P2 substantially triggered the rapid enhancement of haze pollution in NC.

201       In addition to snow cover, soil moisture was another important factor affecting $HD_{NC}$ (Yin and Wang, 2016b). The

September-November soil moisture and $HD_{NC}$ were negatively correlated in central Siberia (54–70°N, 80–130°E; Fig. S3c).
The area-averaged September-November soil moisture over central Siberia was denoted as the Soilw index, whose correlation
coefficients with $HD_{NC}$ were –0.57 and –0.60 from 1979 to 2018 before and after detrending, respectively (above the 99%
confidence level). Negative Soilw anomalies could induce a positive phase of EA/WR, and the associated anticyclonic
circulations occurred more frequently and more strongly (Fig. S7). Correspondingly, the local vertical and horizontal
dispersion conditions were limited. With increasing moisture, pollutants can more easily accumulate in a confined area. The
spatial linear trend of soil moisture also shifted from increasing to decreasing in 2010, opposite to the trend of $HD_{NC}$ (Fig. 3e,
f). The change rate of Soilw was 38.8 mm/10 yr (opposite that of $HD_{NC}$) during P1, and the rate of change became more intense
(–51.8 mm/10 yr) during P2, physically driving a similar large change in $HD_{NC}$ (Fig. 4d).



The varying trends of these four preceding external factors jointly drove the trend reversal of $HD_{NC}$ based on their physical
relationships with the haze pollution in North China. To exclude the impacts of the stage trends of these variables on the
physical links between the climate drivers and $HD_{NC}$, the correlations between these factors and $HD_{NC}$ were explored during
the decreasing stage (i.e., 1979–2010) and increasing stage (2010–2018), and all of these correlations were significant (Table
1). Thus, the physical relationships between $HD_{NC}$ and these four factors were long-standing and did not disappear as the trend
changed. These four external factors had completely opposite trends in P1 and P2. Excluding $SST_A$, the amplitudes of the
change trends of the other three indices in P2 were obviously stronger than those in P1 and were identical to those of $HD_{NC}$
(Table 1). In our study, we composited the simulations based on the GEOS-Chem model to determine the impact on haze
pollution of each factor under the fixed-emissions level. The years in the top 20% and the bottom 20% of the four indices (i.e.,
$SST_P$, $-1 \times SST_A$, Snowc and $-1 \times$Soilw) in P1 and P2 were selected, which could remove the effects of different trends. The
composite differences for the four external forcing factors were significant in the selected regions and passed the Student's t
test (Fig. S8). The responses of simulated $HD_{NC}$ to the original (detrended) sequences of $SST_P$, $SST_A$, Snowc and Soilw were
all positive, which are consistent with the observational results (Fig. 5). Specifically, for the four original (detrended) drivers,
the resulting differences in simulated $HD_{NC}$ were 3.94 (5.28), 5.97 (5.07), 1.86 (1.86) and 6.49 (6.49) days in P1 and 4.46
(4.46), 4.26 (4.26), 7.54 (7.54) and 7.35 (7.35) days in P2 (Fig. 5). These differences were distinct and further confirmed that
each factor played a role in the occurrence of haze pollution in NC.
These four indices were employed to linearly fit $HD_{NC}$ based on a multiple linear regression (MLR) model (Wilks, 2011).
As shown in Figure 4e, the correlation coefficient between the fitted and observed $HD_{NC}$ was 0.82. After a five-year running
average, the correlation coefficient reached 0.92. This model showed good ability to fit the infection point in 2010 and
highlighted the trend changes. Such a good fitting effect indicates that changes in the four external forcing factors could well
have influenced the variation in $HD_{NC}$. By exciting stronger responses in the atmosphere, such as the positive EA/WR phase
and the strengthened anomalous anticyclone over NC, the abovementioned climate drivers created stable and stagnant
environments in which the haze pollution in NC could rapidly exacerbate after 2010 (Table 1). Among the four indices, the
correlation coefficients between $SST_P$ and Snowc (Pair 1) and between $SST_A$ and Soilw (Pair 2) were high, while the rest were
insignificant. The variance inflation factors of the four factors in the MLR model were less than 2, showing that the collinearity
among them was weak. When selecting one factor from both Pair 1 and Pair 2 to refit $HD_{NC}$, the correlation coefficient between
the fitted and observed $HD_{NC}$ and the trends of the fitted $HD_{NC}$ in P2 worsened (Fig. S9). Therefore, these four external factors
were all indispensable to achieve a better fitting effect. The intercorrelated climate factors of Pair 1 and Pair 2 contributed
27.8% and 84.6%, respectively, to the trends of $HD_{NC}$ in P1 and 54.8% and 20.4% to the trends in P2. Thus, the joint effect of
$SST_A$ and Soilw played a more important role in the decreasing trend of $HD_{NC}$ in P1; however, the impacts of $SST_P$ and Snowc
were more than twice those of $SST_A$ and Soilw in P2. More importantly, the fitted curve revealed a decreasing trend of $HD_{NC}$



(–5.24 days/10 yr) that was larger than observed (–4.67 days/10 yr) during P1. Many studies have noted that human activities
have led to persistently increasing trends of $HD_{NC}$ (Yang et al., 2016; Li et al., 2018). The combination of the exorbitant
decreased trend indicated by climate conditions and the long-term trend from anthropogenic emissions resulted in a realistic
slow decline (Table 2). This proportion of the trend explained by climate drivers (72.3%) decreased in P2 because the
increasing trend driven by the climate divers and emissions jointly led to a rapid increase in $HD_{NC}$.
**5 Conclusions and discussions**
Haze events in early winter in North China exhibited rapid growth after 2010, which was completely different from the
slow decline observed before 2010, showing a trend reversal in the year 2010 (Fig. 1). The trend changes of associated
meteorological conditions exhibited identical responses. After 2010, the lower BLH, weakened wind speed, sufficient moisture
and anomalous ascending motion (all with larger tendencies than before 2010) limited the horizontal and vertical dispersion
conditions and thus enhanced the occurrence of early winter haze pollution (Fig. 2). However, before 2010, the climate
conditions showed the opposite characteristics and could create an environment with adequate ventilation for the dissipation
of particles.
In this study, the external forcing factors that caused the significant growth of $HD_{NC}$ after 2010 and the associated physical
mechanisms were investigated. These factors could stimulate and strengthen the anomalous anticyclone over NC via exciting
the EA/WR teleconnection pattern, thus regulating the meteorological conditions, weakening the dispersion conditions and
facilitating the accumulation of haze pollutants. The four climate drivers physically related to $HD_{NC}$ showed exactly opposite
trend changes with an inflection point in 2010, which agrees with the behavior of $HD_{NC}$ (Fig. 4). The factors of Pair 1 ($SST_A$
and Soilw) and Pair 2 ($SST_P$ and Snowc) had joint effects and played more important roles in the increasing trend of $HD_{NC}$ in
P2 and the decreasing trend of $HD_{NC}$ in P1, respectively (Table 2). The fitting result of the four factors with the trend of $HD_{NC}$
showed a strongly decreasing trend in P1 and a weakly increasing trend in P2. Together with increasing emissions, these factors
jointly led to a relatively slow decreasing trend of $HD_{NC}$ before 2010 and rapid growth afterward. Therefore, both the
decreasing trend in P1 and the increasing trend in P2 were caused by a combination of climate drivers and emissions.
Anthropogenic emissions have exceeded a high level since the 1990s, providing a sufficient foundation for the generation
of severe haze pollution (Fig. 1). However, the effects of climate variability delayed warnings before 2010. Together with the
local meteorological conditions, the trends of the climate drivers reversed in 2010, initiating a dramatically increase in $HD_{NC}$
after 2010, which quickened the worsening of haze pollution in NC (Fig. 5; Table 1). The superimposed effect of high-level
human emissions with strengthened climate anomalies loudly sounded the alarms through the extremely rapid rise of haze
pollution.



*Data availability.* The monthly mean meteorological data are obtained from NCEP/NCAR reanalysis datasets
(https://www.esrl.noaa.gov/psd/data/gridded/data.ncep.reanalysis.html). The boundary layer height data are available from the
Interim reanalysis dataset (http://www.ecmwf.int/en/research/climate-reanalysis/era-interim). The number of haze days can be
obtained from the authors. The $PM_{2.5}$ concentrations from 2009 to 2016 can be downloaded from the US embassy
(http://www.stateair.net/web/historical/1/1.html), and those from 2014 to 2018 can be downloaded from China National
Environmental Monitoring Centre (http://beijingair.sinaapp.com/). The monthly total emissions of BC, $NH_3$, $NO_x$, OC, $SO_2$,
$PM_{10}$ and $PM_{2.5}$ are obtained from the Peking University emission inventory (http://inventory.pku.edu.cn/). SST data are
downloaded from http://www.esrl.noaa.gov/psd/data/gridded/data.noaa.ersst.v4.html. Soil moisture data are obtained from
https://www.esrl.noaa.gov/psd/data/gridded/data.cpcsoil.html. Snow cover data can be downloaded from Rutgers University:
http://climate.rutgers.edu/snowcover/. The emissions of 2010 can be downloaded from
http://geoschemdata.computecanada.ca/ExtData/HEMCO/MIX.


**Acknowledgements**
This work was supported by the National Key Research and Development Plan (2016YFA0600703), National Natural Science
Foundation of China (41705058, 41991283 and 91744311), and the funding of Jiangsu innovation & entrepreneurship team.
**Author contributions**
Wang H. J. and Yin Z. C. designed the research. Yin Z. C. and Zhang Y. J. performed research. Zhang Y. J. prepared the
manuscript with contributions from all co-authors.
**Competing interests**
The authors declare no conflict of interest.






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



**Table and Figure legends**
**Table 1.** Correlation coefficients (CCs) between $HD_{NC}$ and the $SST_P$, $SST_A$, Snowc and Soilw indices after detrending and the
trends of the $SST_P$, $SST_A$, Snowc and Soilw indices for the periods 1991–2010 and 2010–2018. $CC_1$, $CC_2$, and $CC_3$ represent
the correlation coefficients from 1979 to 2018, 1979 to 2010 and 2010 to 2018, respectively. "***" indicates that the CC was
above the 99% confidence level, "**" indicates that the CC was above the 95% confidence level, and "*" indicates that the
CC was above the 90% confidence level.
**Table 2.** The contribution rate of fitted $HD_{NC}$ and each external forcing factor to the trend of $HD_{NC}$ in P1 and P2, respectively.
**Figure 1.** (a) Variations in the December-January emissions (unit: Tg) of black carbon (BC), ammonia ($NH_3$), nitrogen oxide
($NO_x$), organic carbon (OC), sulfur dioxide ($SO_2$), $PM_{10}$ and $PM_{2.5}$ over North China from 1979 to 2013 and the variation in
$HD_{NC}$ from 1979 to 2018 (black solid line). The red dashed line represents the total emissions of the seven pollutants. The blue
and green solid (dashed) lines indicate the number of days when the hourly $PM_{2.5}$ concentrations in a day exceeded 75 μg m$^{-3}$
and 100 μg m$^{-3}$, respectively, from 2009 to 2016 (2014 to 2018) using observed data from the US embassy (China National
Environmental Monitoring Centre). (b) Temporal evolutions of $HD_{NC}$ (in black), simulated haze days (unit: days; red) and (c)
average $PM_{2.5}$ concentrations (unit: μg m$^{-3}$; blue) in NC. The dashed lines denote linear regressions for 1991–2010 (P1) and
2010–2018 (P2). Trend 1 and Trend 2 represent the linear trends of the simulations in P1 and P2, respectively.
**Figure 2.** Area-averaged linear trends of the BLH (unit: m/yr), specific humidity (unit: %/10 yr), surface wind speed (unit: m
s$^{-1}$/10$^2$ yr) and omega (unit: pascal s$^{-1}$/10$^3$ yr) over NC in early winter for the periods 1991–2010 (P1) and 2010–2018 (P2).
All datasets were 5-year running averages before calculating the trends.
**Figure 3.** Linear trends of the Pacific and Atlantic SST (unit: °C/yr; a, b), Eurasian snow cover (unit: %/yr; c, d), and central
Siberian soil moisture (unit: mm/yr; e, f) for the periods 1991–2010 (P1) and 2010–2018 (P2). All datasets were 5-year running
averages before calculating the trends. The green boxes represent the regions where the four indices are defined. Black dots
indicate that the trends were above the 95% confidence level.
**Figure 4.** Variations in $HD_{NC}$ (in black) and the $SST_P$ (unit: °C; a, red), $SST_A$ (unit: °C; b, blue), Snowc (unit: %; c, yellow),
and Soilw (unit: mm; d, green) indices and the $HD_{NC}$ values fitted by the MLR model by the above four factors (unit: days; e,
purple) from 1979 to 2018. Thick lines indicate 5-year running averaged time series. The rectangles and triangles indicate the
inflection points of $HD_{NC}$ and the four indices, which were tested by the Mann-Kendall test.
**Figure 5.** Composite of the simulated $HD_{NC}$ caused by the four external forcing factors (Favor Years minus Unfavor Years).
The circles and crosses represent the original and detrended sequences, respectively.





**Table 1.** Correlation coefficients (CCs) between $HD_{NC}$ and the $SST_P$, $SST_A$, Snowc and Soilw indices after detrending and the trends of the $SST_P$, $SST_A$, Snowc and Soilw indices for the periods 1991–2010 and 2010–2018. $CC_1$, $CC_2$, and $CC_3$ represent the correlation coefficients from 1979 to 2018, 1979 to 2010 and 2010 to 2018, respectively. "***" indicates that the CC was above the 99% confidence level, "**" indicates that the CC was above the 95% confidence level, and "*" indicates that the CC was above the 90% confidence level.

| | CC with $HD_{NC}$ | Trend / 10yr | |
|---|---|---|---|
| | | 1991–2010 | 2010–2018 |
| $SST_P$ | $CC_1 = 0.67$ *** | | |
| | $CC_2 = 0.39$ ** | −0.20 °C*** | 1.99 °C*** |
| | $CC_3 = 0.66$ *** | | |
| $SST_A$ | $CC_1 = -0.64$ *** | | |
| | $CC_2 = -0.54$ *** | 0.55 °C*** | −0.52 °C*** |
| | $CC_3 = -0.61$ *** | | |
| Snowc | $CC_1 = 0.54$ *** | | |
| | $CC_2 = 0.46$ *** | −1.79%** | 28.35%*** |
| | $CC_3 = 0.53$ *** | | |
| Soilw | $CC_1 = -0.60$ *** | | |
| | $CC_2 = -0.30$ * | 38.78mm*** | −51.81mm*** |
| | $CC_3 = -0.66$ *** | | |

**Table 2.** The contribution rate of fitted $HD_{NC}$ and each external forcing factor to the trend of $HD_{NC}$ in P1 and P2, respectively.

| | Fitted $HD_{NC}$ | $SST_P$ | $SST_A$ | Snowc | Soilw |
|---|---|---|---|---|---|
| P1 | 112.2% | 23.3% | 43.9% | 4.5% | 40.7% |
| P2 | 72.3% | 41.9% | 7.5% | 12.9% | 10.0% |

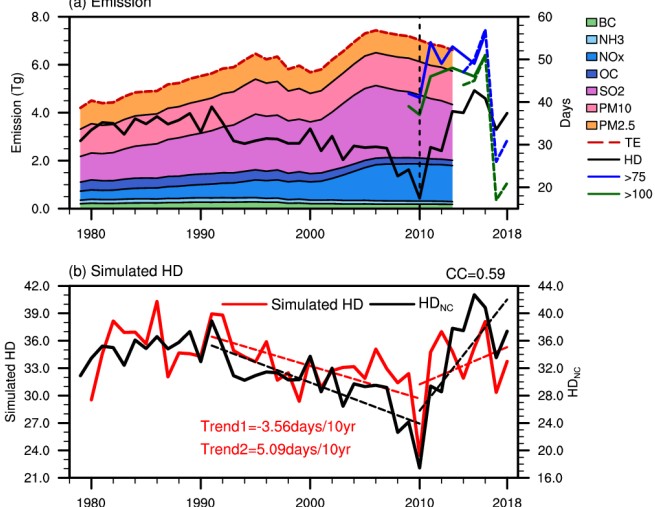

466

**Figure 1.** (a) Variations in the December-January emissions (unit: Tg) of black carbon (BC), ammonia ($NH_3$), nitrogen oxide ($NO_x$), organic carbon (OC), sulfur dioxide ($SO_2$), $PM_{10}$ and $PM_{2.5}$ over North China from 1979 to 2013 and the variation in $HD_{NC}$ from 1979 to 2018 (black solid line). The red dashed line represents the total emissions of the seven pollutants. The blue and green solid (dashed) lines indicate the number of days when the hourly $PM_{2.5}$ concentrations in a day exceeded 75 µg m$^{-3}$ and 100 µg m$^{-3}$, respectively, from 2009 to 2016 (2014 to 2018) using observed data from the US embassy (China National Environmental Monitoring Centre). (b) Temporal evolutions of $HD_{NC}$ (in black), simulated haze days (unit: days; red) and (c) average $PM_{2.5}$ concentrations (unit: µg m$^{-3}$; blue) in NC. The dashed lines denote linear regressions for 1991–2010 (P1) and 2010–2018 (P2). Trend 1 and Trend 2 represent the linear trends of the simulations in P1 and P2, respectively.


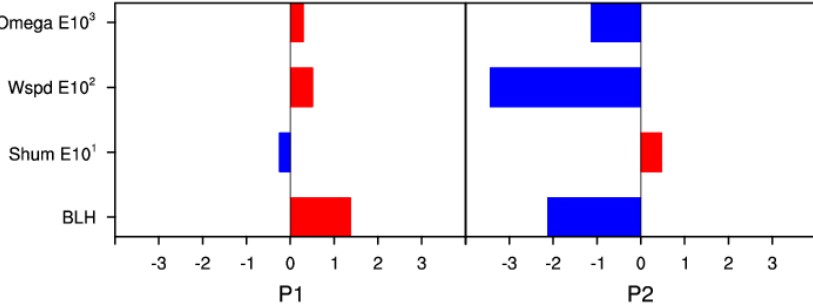


**Figure 2.** Area-averaged linear trends of the BLH (unit: m/yr), specific humidity (unit: %/10 yr), surface wind speed (unit: m s$^{-1}$/10$^2$ yr) and omega (unit: pascal s$^{-1}$/10$^3$ yr) over NC in early winter for the periods 1991–2010 (P1) and 2010–2018 (P2). All datasets were 5-year running averages before calculating the trends.






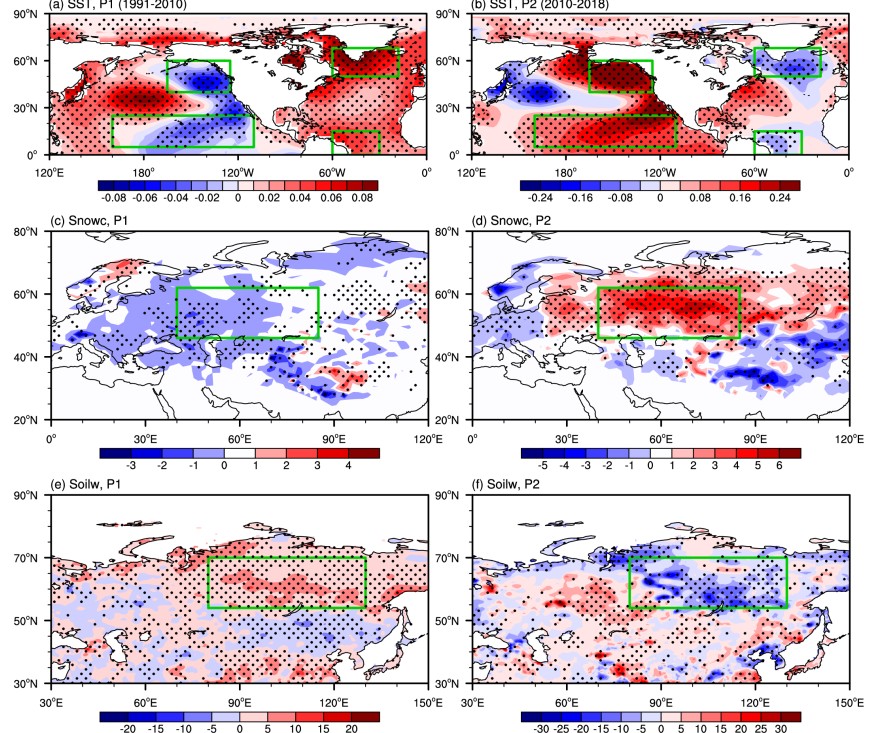


**Figure 3.** Linear trends of the Pacific and Atlantic SST (unit: °C/yr; a, b), Eurasian snow cover (unit: %/yr; c, d), and central

Siberian soil moisture (unit: mm/yr; e, f) for the periods 1991–2010 (P1) and 2010–2018 (P2). All datasets were 5-year

running averages before calculating the trends. The green boxes represent the regions where the four indices are defined.

Black dots indicate that the trends were above the 95% confidence level.

486
487



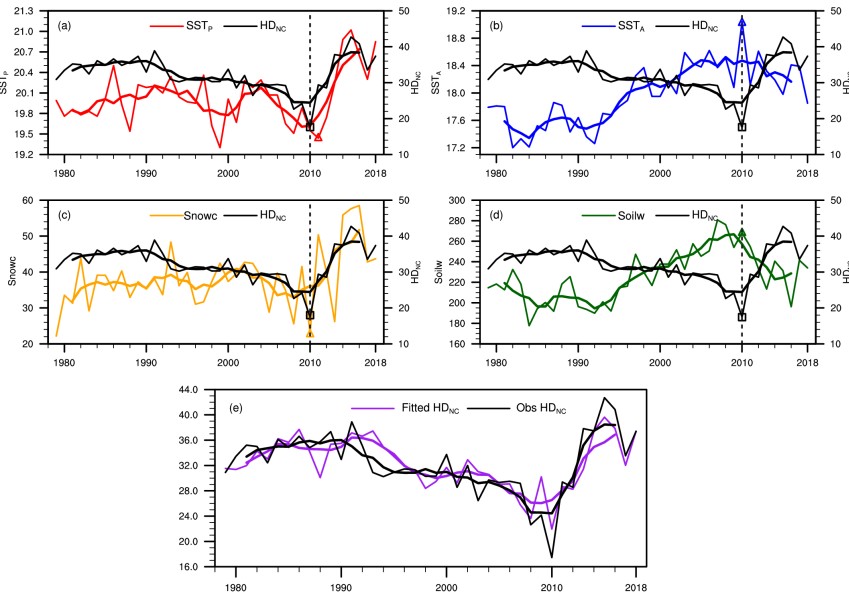

488

**Figure 4.** Variations in $HD_{NC}$ (in black) and the $SST_P$ (unit: °C; a, red), $SST_A$ (unit: °C; b, blue), Snowc (unit: %; c, yellow),

and Soilw (unit: mm; d, green) indices and the $HD_{NC}$ values fitted by the MLR model by the above four factors (unit: days;

e, purple) from 1979 to 2018. Thick lines indicate 5-year running averaged time series. The rectangles and triangles indicate

the inflection points of $HD_{NC}$ and the four indices, which were tested by the Mann-Kendall test.

493

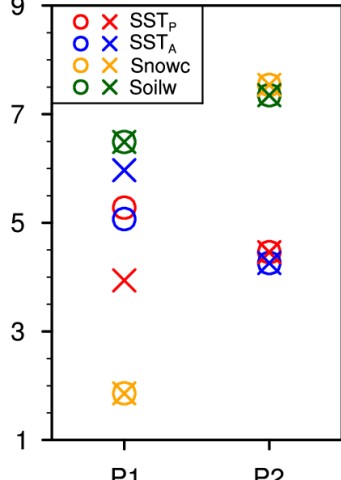

494

**Figure 5.** Composite of the simulated $HD_{NC}$ caused by the four external forcing factors (Favor Years minus Unfavor Years).

The circles and crosses represent the original and detrended sequences, respectively.

497