# Peer review of "Roles of Climate Variability on the Rapid Increases of Early Winter Haze Pollution in North China after 2010"

_Atmospheric Chemistry and Physics, 2020_

## Referee Comment (RC1) · Shaw Chen Liu (Referee) · 1 Jul 2020

General comments: This paper presents results from a study on the impact of climate variability on the rapid increase of winter haze pollution in northern China around 2011-2015. It is a well written paper on an important subject. The authors have posed a pivotal scientific question about the haze pollution in northern China, which can have a critical implication on the long-term trends of haze. In addition, the authors have addressed the scientific question effectively by using a nested-grid global photochemical model. I believe that the paper deserves publication in ACP provided that the following two specific comments are addressed.

Specific comments: 1. The main argument of the paper depends fundamentally on the validity of model simulations described in the "section 2.2 Geos-Chem description and experimental design". It is essential that the performance of the model used to simulate haze pollution in North China is validated or at least evaluated against observations. Figure 1 of the paper could be used to some extent for evaluating the model performance, but additional simulations with historical emissions are needed. Judging from the trends of anthropogenic emissions, this reviewer is afraid that results from simulations with historical emissions might turn out to be significantly different from observed haze days. In any case, uncertainty in the model needs to be included in the discussions of sections 3-5. 2. "The autumn SST in the Pacific and Atlantic, Eurasian snow cover and central Siberian soil moisture, which exhibited completely opposite trends before and after 2010, were proven to stimulate identical trends of meteorological conditions related to haze pollution in North China." in the abstract and conclusion section maybe an overstatement, at least in terms of the relatively large uncertainty of the model. A more fundamental concern is that the method used in evaluating contributions of the four climate drivers does not imply any causal relationship.

---

## Referee Comment (RC2) · Anonymous Referee #2 · 22 Jul 2020

The authors proposed a very interesting question that haze pollution in early winter in North China experienced a rapid increase from 2010 after a two-decade decrease. This notion is supported by long-term haze days defined by visibility and relative humidity. By using model simulation and statistical analysis, they argued that climate variability is the dominant driver for rapid increase after 2010. They further analyzed the possible external climate forcings to support their conclusions. Overall, the topic of this study fits this journal well. The authors conducted modeling and statistical efforts to defend their conclusions. I think it is publishable before some concerns in the following are addressed.
1. I am most concerned about the rapid increase trend after 2010, because it looks like the trend is mainly driven by an extreme anomaly in 2010. As for visibility-based haze days, year 2010 doesn't change the increasing trend. But, I think the increasing trend will not hold for observed PM2.5 and simulated haze days if year 2010 is removed. If this is the case, that means the authors should take caution stating that there is rapid increase of haze pollution after 2010. It is better to focus on the long-term trend than only highlight the rapid increase after 2010. Other PM2.5-related measurements (e.g., satellite AOD) might be helpful.

2. It is surprised that the simulations with fixed anthropogenic emissions can produce the observed reversal frequency of haze days very well. You also cite Dang and Liao (2019) to support this argument. I take a look at this reference, but I think their results didn't show an increasing trend after 2010. I am very curious about if the role of anthropogenic emissions is very limited after 2010.

Specific comments:

Title: The authors focused on "early winter" in this study, but you failed to define what's "early winter", December and January? Please clarify this in the title and also in the main text. Also, I am not sure if the conclusions support the "rapid increase" after 2010.

Line19: You mentioned "human emissions" started to decline in mid-2000s. How did they jointly lead to rapid increase in haze days?

Line91: MLR should spell out.

Line 102: The trend should be given with its statistical significance. Please check throughout the text.

Lines 109-110: Only PM2.5 data in Beijing used from the national measurement network? Or PM2.5 over North China? This should be also clarified in the caption of Figure 1.

Lines 136-137: I think they also show results with varying meteorology?

**ACPD**
Figure 1a: missing legend for green dash line. I don't think it is reasonable to show the red dash line. The sum of these pollutant emissions doesn't make sense.

Figure 1b: You should give observed trends of haze days along with the simulated trends.

---

## Author Comment (AC1) · 17 Aug 2020

**Reply to Reviewer #1:**

General comments: This paper presents results from a study on the impact of climate variability on the rapid increase of winter haze pollution in northern China around 2011-2015. It is a well written paper on an important subject. The authors have posed a pivotal scientific question about the haze pollution in northern China, which can have a critical implication on the long-term trends of haze. In addition, the authors have addressed the scientific question effectively by using a nested-grid global photochemical model. **I believe that the paper deserves publication in ACP** provided that the following two specific comments are addressed.

1.  **(1) The main argument of the paper depends fundamentally on the validity of model simulations described in the "section 2.2 Geos-Chem description and experimental design". It is essential that the performance of the model used to simulate haze pollution in North China is validated or at least evaluated against observations. Figure 1 of the paper could be used to some extent for evaluating the model performance, but additional simulations with historical emissions are needed.**
    **(2) Judging from the trends of anthropogenic emissions, this reviewer is afraid that results from simulations with historical emissions might turn out to be significantly different from observed haze days. In any case, uncertainty in the model needs to be included in the discussions of sections 3-5.**

*Reply:*

(1) We selected the year of 2015, which has just begun to strengthen emission reduction, and 2017, which has launched the air pollution prevention and management plan for "2+26" cities (Yin and Zhang, 2020), as two representative years to **simulate the actual PM$_{2.5}$ concentration**, so as to evaluate the performance of the GEOS-Chem model. The emission factors and meteorological conditions of 2015 and 2017 were be used respectively to simulate the PM$_{2.5}$ concentrations in early winters of 2015 and 2017. The simulation results are very **close to the observed data** in the two years (Figure S3) with high correlation coefficients reaching **0.88 and 0.85**, indicating that **the simulated data could basically reflect the change of actual PM$_{2.5}$ concentrations.** We have added this part about the model evaluation in the Section 2.2, and added a new Figure S3.

[Figure]

**Figure S3.** Temporal evolutions of observed (black) and simulated (red) PM2.5 concentrations (unit: μg m$^{-3}$; blue) in 2015 (a) and 2017 (b) early winter in North China.

In fact, the GEOS-Chem model has a **wide application**, and we have introduced **a few applications of others studies** in Section 2.2 to demonstrate the performance of the model laterally.

(2) Our simulations were designed to **emphasize the effects of climate**, so we used fixed emissions. In addition to the experiments with fixed emission in 2010, **a new set of experiments** was carried out by GEOS-Chem with **fixed emissions in 1985**, representing a low emission level. This simulation of the frequency of haze days (>50 μg m$^{-3}$) also **reproduced the trend reversal of haze pollution very well (Figure R1), similarly with the observed haze days.**

[Figure]

**Figure R1.** Temporal evolutions of HD$_{NC}$ (in black), simulated haze days under 1985 (unit: days; blue) emission scenarios in NC. The dashed lines denote linear regressions for 1991–2010 (P1) and

2010–2018 (P2). The black and blue Trend 1 and Trend 2 represent the linear trends of the observed and simulated haze days 1985 emission scenarios in P1 and P2, respectively.

Considering the change of anthropogenic emission, in Dang and Liao's finding (2019), the CRTL experiment (black line in Figure R2a) also **showed a decreased trend during 1985 to 2002** (blue grids in Figure R2b) and **an increased trend after 2010** (red grids in Figure R2b). Although this trend is weaker than the MET experiment, which only considered the effects of meteorological conditions (green line R2a), the overall change of trend in the CTRL experiment is **consistent with observations**. And Mao L. et al. (2019, National Science Review) also raised the **contradiction between the change of anthropogenic emissions and haze days** based on an observation approach, which was manifested as the number of winter haze days with no significant trend in most provinces and districts in eastern China from 1973 to 2012, contrary to the 2.5-fold increase in the emissions of particulate matter and its precursors (PM emissions) in the same period of time. And we have **added the discussion of the uncertainties** in section 5.

[Figure]

1. CTRL is the control simulation with variations in meteorological parameters, anthropogenic emissions, and biomass burning emissions from 1985 to 2017.

2. EMIS is the simulation with changes in anthropogenic and biomass burning emissions over 1985–2017, while the meteorological fields were fixed at the 1985 levels. The aim of this simulation is to quantify the impacts of changes in emissions on SWHDs during 1985–2017.

3. MET is the simulation with changes in meteorological fields over 1985–2017, while anthropogenic and biomass burning emissions were fixed at the 2015 levels. This simulation is set to examine the impacts of changes in meteorological parameters on SWHDs during 1985–2017.

**Figure 10.** (a) Time series of frequencies (days) of regional SWHD in BTH from three simulations (CTRL, EMIS, MET) for 1985 to 2017. (b–e) Time series of linear trends calculated over different periods for simulated frequencies of the (b) CTRL, (c) EMIS, and (d) MET simulations. The x axis indicates the start year, and the y axis indicates the number of years since the start year during which period the trend is calculated. The filled color in each square shows the calculated trend value, and those values marked with black borders are significant above the 95 % confidence level.

**Figure R2**. A key figure and design of three numerical experiments in *Dang and Liao (2019) published in Atmos. Chem. Phys.*

***Related references:***

Yin, Z., and Zhang, Y.: Climate anomalies contributed to the rebound of PM2.5 in winter 2018 under intensified regional air pollution preventions, Sci. Total Environ.,

726, 138514, 2020.

Dang, R. and Liao, H.: Severe winter haze days in the Beijing–Tianjin–Hebei region from 1985 to 2017 and the roles of anthropogenic emissions and meteorology, Atmos. Chem. Phys., 19, 10801–10816, https://doi.org/10.5194/acp-19-10801-2019, 2019.

Mao, L., Liu, R., Liao, W., Wang, X., Shao, M., Liu, S. and Zhang, Y.: An observation-based perspective of winter haze days in four major polluted regions of China. National Science Review, **6**, 515–523, doi: 10.1093/nsr/nwy118, 2019.

*Revisions:*

**Lines 79-85:** At present, GEOS-Chem model has been widely used, Dang et al. (2019) showed that the simulated spatial patterns and daily variations of winter $PM_{2.5}$ based on this model agree well with the observations from 2013 to 2017, available years with measured $PM_{2.5}$. We selected the year of 2015, which has just begun to strengthen emission reduction, and 2017, which has launched the air pollution prevention and management plan for "2+26" cities (Yin and Zhang, 2020), as two representative years to simulate the actual $PM_{2.5}$ concentrations, so as to evaluate the performance of the GEOS-Chem model. The simulation results are very close to the observed data (Fig. S3) with high correlation coefficients reaching 0.88 and 0.85 in 2015 and 2017, indicating this model could basically reflect the change of actual $PM_{2.5}$ concentrations.

**Lines 278-282:** Note that a number of factors contribute to the uncertainties in our results. Although a high emission scenario was used to simulate the number of haze days and emphasized the influence of meteorology, no complete and varied emission inventories were used to drive the GEOS-Chem model to make a comparison, which caused certain uncertainty. Furthermore, when assessing the contribution percentages of the external forcing factors, the coupling effect between climate variability and anthropogenic emissions was not considered, therefore the contribution rate of climate conditions might be overestimated.

**2. "The autumn SST in the Pacific and Atlantic, Eurasian snow cover and central Siberian soil moisture, which exhibited completely opposite trends before and after 2010, were proven to stimulate identical trends of meteorological conditions related to haze pollution in North China." in the abstract and conclusion section**

**maybe an overstatement, at least in terms of the relatively large uncertainty of the model. A more fundamental concern is that the method used in evaluating contributions of the four climate drivers does not imply any causal relationship.**

*Reply:*

We have adopted **a more rigorous statement** to explain the effect of the four external forcing factors on haze events in abstract and conclusion section.

*Revisions:*

**Lines 14-16:** The autumn SST in the Pacific and Atlantic, Eurasian snow cover and central Siberian soil moisture, which exhibited completely opposite trends before and after 2010, might had close relationships with the identical trends of meteorological conditions related to haze pollution in North China …….

**Lines 267-269:** In this study, the external forcing factors that closely related to the significant growth of $HD_{NC}$ after 2010 and the associated physical mechanisms were investigated. These factors might strongly linked to the anomalous anticyclone over NC via exciting the EA/WR teleconnection pattern……

---

## Author Comment (AC2) · 17 Aug 2020

**Reply to Reviewer #2:**

General comments: The authors proposed a very interesting question that haze pollution in early winter in North China experienced a rapid increase from 2010 after a two-decade decrease. This notion is supported by long-term haze days defined by visibility and relative humidity. By using model simulation and statistical analysis, they argued that climate variability is the dominant driver for rapid increase after 2010. They further analyzed the possible external climate forcings to support their conclusions. Overall, **the topic of this study fits this journal well**. The authors conducted modeling and statistical efforts to defend their conclusions. **I think it is publishable** before some concerns in the following are addressed.

**1. I am most concerned about the rapid increase trend after 2010, because it looks like the trend is mainly driven by an extreme anomaly in 2010. As for visibility-based haze days, year 2010 doesn't change the increasing trend. But, I think the increasing trend will not hold for observed PM2.5 and simulated haze days if year 2010 is removed. If this is the case, that means the authors should take caution stating that there is rapid increase of haze pollution after 2010. It is better to focus on the long-term trend than only highlight the rapid increase after 2010. Other PM2.5-related measurements (e.g., satellite AOD) might be helpful.**

*Reply:*

Many recent studies have focused on the **long-term trend in the haze problem and shown that might been driven by human activities and global warming** (Li et al, 2018; Yang et al, 2016; Horton et al, 2014; Cai et al., 2017). However, none of the above studies focused on the change in the haze trend. Therefore, our study is to novelly **focus on the trend reversal of HD$_{NC}$ around 2010, which had stage inconsistencies with the trend of emissions and global warning**, and to explain the reasons from the perspective of climate change, which is the innovation of this work.

(1) We **excluded the extreme anomaly in 2010** and selected two periods from 1991 to 2009 and from 2011 to 2018 respectively to focus on the trend changes, so as to **further confirm the existence of the rapid increase after 2010**. We can be sure that the **trend reversal still exists and has not changed by removing year 2010**. In the two periods removing 2010, HD$_{NC}$ also showed slowly decreased during 1991–2009

(P1) with a rate of **3.82 days/10 yr** but rapidly increased during 2011–2018 (P2) with a rate of **20.76 days/10 yr**, and both passed the 95% t test.

(2) The number of haze days calculated **using observed data** in Beijing from the US embassy (available from 2009 to 2016) also showed **an increasing trend without 2010** (Figure R1), with a rate of 2.3days/10yr and 6.0days/10yr when exceeding 75 µg m$^{-3}$ and 100 µg m$^{-3}$, respectively. The AOD data is monthly average, so it could not calculate the number of polluted days and cannot make comparison with HD$_{NC}$.

[Figure]

**Figure R1.** The number of days when the hourly PM$_{2.5}$ concentrations in a day exceeded 75 µg m$^{-3}$ (blue) and 100 µg m$^{-3}$ (green), respectively, from 2009 to 2016 using Beijing observed data from the US embassy. The dash lines represent the trends during 2010-2016.

(3) We also check a lot of previous studies, and can show you many **hard evidences** that the **trend reversal is reliable**. Many **observation-based researches also showed the same trend as ours**. Here, I summarized a part of them (independent researches and irrelevant with us) to show the **consistency** with our results. Actually, recent studies generally revealed that the boreal winter haze days across North China **had a trend reversal in 2010**, which is consistent with the result in our research (Shi et al., 2019; Mao et al., 2019). As the most polluted area in North China, Beijing-Tianjing-Hebei (BTH) region had **a statistically significant decline trend** of haze days during 1990-2010 (Figure R2; purple arrow), and **a rapid increase trend after 2010** (Figures R2; red arrow).

[Figure]

**Figure R2**. Variation of HD revealed by other researchers, the purple arrows indicate the trend approximately from 1990 to 2010, and the red arrows indicate the trend after 2010. (a) Mean winter haze days (black dashed lines) and visibility (red dashed lines) in the Beijing-Tianjin-Hebei (BTH) region (the most polluted area in North China) for 1961–2014. A 15-year low-pass Gaussian filter is drawn with solid lines. ***This panel (a) was extracted from Shi et al., (2019) published on Atmospheric Research***. (b) HD in North China during the period 1973-2016. Absolute values are shown in blue, detrended values in red. Dashed lines denote linear regressions. ***This panel (b) was extracted from Mao et al., (2019) published on National Science Review***.

(4) In 2017, the number of haze days decreased significantly, which was caused by the **enhanced "2+26" emission reduction measures and the cold winter**. However, this **did not affect the increased trend** after 2010. And in 2018, due to adverse meteorological conditions, even under strong emission reduction, PM$_{2.5}$ concentration had a significant rebound (Yin and Zhang, 2020).

*Related References:*

Li, K., Liao, H., Cai, W., Yang, Y.: Attribution of anthropogenic influence on atmospheric patterns conducive to recent most severe haze over eastern China, Geophys. Res. Lett., 45(4), 2072-2081, doi:10.1002/2017GL076570, 2018.

Yang, Y., Liao, H., Lou, S.: Increase in winter haze over eastern China in recent decades: Roles of variations in meteorological parameters and anthropogenic emissions, J. Geophys. Res. Atmos., 121: 13050–13065, 2016.

Horton, D., Skinner, C., Singh, D., Diffenbaugh, N.: Occurrence and persistence of future atmospheric stagnation events, Nat. Clim. Change, 4, 698–703, 2014.

Cai, W., Li, K., Liao, H., Wang, H., and Wu, L.: Weather conditions conducive to Beijing severe haze more frequent under climate change, Nat Clim Change, 7, 257–262,

2017.

Shi, P., Zhang, G., Kong, F., Chen, D., Cesar Azorin-Molina, Jose A. Guijarro: Variability of winter haze over the Beijing-Tianjin-Hebei region tied to wind speed in the lower troposphere and particulate sources. Atmospheric Research, **215**, 1–11, 2019.

Mao, L., Liu, R., Liao, W., Wang, X., Shao, M., Liu, S. and Zhang, Y.: An observation-based perspective of winter haze days in four major polluted regions of China. National Science Review, **6**, 515–523, doi: 10.1093/nsr/nwy118, 2019.

Yin, Z., and Zhang, Y.: Climate anomalies contributed to the rebound of PM2.5 in winter 2018 under intensified regional air pollution preventions, Sci. Total Environ., 726, 138514, 2020.

***Revisions:***

**Lines 112-114:** Excluding year 2010 did not affect the change in the trend of the two periods, with a decreased rate of 3.82 days/10 yr during 1991–2009, and an increased rate of 20.76 days/10 yr during 2011–2018 (passing 95% t test)…….

**2. It is surprised that the simulations with fixed anthropogenic emissions can produce the observed reversal frequency of haze days very well. You also cite Dang and Liao (2019) to support this argument. I take a look at this reference, but I think their results didn't show an increasing trend after 2010.**
**I am very curious about if the role of anthropogenic emissions is very limited after 2010.**

***Reply:***

In addition to the experiments with fixed emission in 2010, **a new set of experiments** was carried out by GEOS-Chem to **further enhance the reliability of the simulation**. The new simulation had changing meteorological fields in winter from 1980 to 2018 but the **fixed emissions in 1985** representing a low emission level. This simulation of the frequency of haze days ($>50$ µg m$^{-3}$) also **reproduced the trend reversal of haze pollution very well** (Figure R3). The simulation results are highly correlated with HD$_{NC}$ and show a rapid increase after 2010. Therefore, the simulation results **can well reproduce the trend reversal of haze days** in both high and low emission levels, indicating that the **simulations are reliable and the rapid growth of haze after 2010 exists.**

[Figure]

**Figure R3.** Temporal evolutions of HD$_{NC}$ (in black), simulated haze days under 1985 (unit: days; blue) emission scenarios in NC. The dashed lines denote linear regressions for 1991–2010 (P1) and 2010–2018 (P2). The black and blue Trend 1 and Trend 2 represent the linear trends of the observed and simulated haze days 1985 emission scenarios in P1 and P2, respectively.

In Dang and Liao's study, the definition of serious haze day is a day with daily mean PM$_{2.5}$ concentration exceeding **150 µg m$^{-3}$**, but in our study, it is exceeding **75 µg m$^{-3}$**. Therefore, we differ in the results of the change of trend. In the MET experiment, the increased trend is clear (Figure R4a, green line and red grid). This increasing trend **is more pronounced through a 9-year weighted moving average method** (Figure R4b). When considering changes in anthropogenic emission, that is, CTRL experiments, the **decreased trend during 1990-2010 was weaker** than in the MET experiment, which only considered the effect of meteorology. On the basis of the meteorological effect that caused haze decreased in P1 and haze increased in P2, a **continuous rising effect of emission** was superimposed. Both of them jointly lead to a **weakened decreased in P1 and a rapid increase in P2**. When assessing the contribution percentages of the external forcing factors, **the coupling effect between climate and emissions was not considered**, therefore the contribution rate of climate conditions might be **overestimated**. We have **added the discussion of this uncertainty** in section 5. For the long-term trend of haze, human activities are the **recognized and fundamental driver** (Li et al., 2018; Yang et al 2016). In our study, we focused on the trend reversal of **early stage decrease and later increase**, and explained it from a climate perspective.

[Figure]

[Figure]

**Figure 10. (a)** Time series of frequencies (days) of regional SWHD in BTH from three simulations (CTRL, EMIS, MET) for 1985 to 2017. **(b–e)** Time series of linear trends calculated over different periods for simulated frequencies of the **(b)** CTRL, **(c)** EMIS, and **(d)** MET simulations. The *x* axis indicates the start year, and the *y* axis indicates the number of years since the start year during which period the trend is calculated. The filled color in each square shows the calculated trend value, and those values marked with black borders are significant above the 95 % confidence level.

**Figure R4**. Key figures in ***Dang and Liao (2019) published in Atmos. Chem. Phys***, including Figure 10 and 12.

*Related references:*

Dang, R. and Liao, H.: Severe winter haze days in the Beijing–Tianjin–Hebei region from 1985 to 2017 and the roles of anthropogenic emissions and meteorology, Atmos. Chem. Phys., 19, 10801–10816, https://doi.org/10.5194/acp-19-10801-2019, 2019.

Li, K., Liao, H., Cai, W., Yang, Y.: Attribution of anthropogenic influence on atmospheric patterns conducive to recent most severe haze over eastern China, Geophys. Res. Lett., 45(4), 2072-2081, doi:10.1002/2017GL076570, 2018.

Yang, Y., Liao, H., Lou, S.: Increase in winter haze over eastern China in recent decades: Roles of variations in meteorological parameters and anthropogenic emissions, J. Geophys. Res. Atmos., 121: 13050–13065, 2016.

*Revisions:*

**Lines 281-282:** Furthermore, when assessing the contribution percentages of the external forcing factors, the coupling effect between climate and emissions was not considered, therefore the contribution rate of climate conditions might be overestimated.

**Line 283:** For the long-term trend of haze, human activities are the recognized and fundamental driver (Li et al., 2018; Yang et al 2016).

**Specific comments:**

**Title: The authors focused on "early winter" in this study, but you failed to define what's "early winter", December and January? Please clarify this in the title and also in the main text. Also, I am not sure if the conclusions support the "rapid increase" after 2010.**

*Reply:*

In our study, the early winter is mean December and January. We further clarify that we focus on the trend change of haze in early winter in the title, and added the definition of early winter in the main text.

*Revisions:*

**Line 1:** Roles of Climate Variability on the Rapid Increases of Early Winter Haze Pollution in North China after 2010

**Line 11:** ……The number of haze days in early winter (December and January) in North China increased rapidly after 2010 but declined slowly before 2010,……

**Line 107:** ……The number of haze days in early winter (December and January) in North China ($HD_{NC}$) reached a remarkable inflection point in 2010……

**Line19: You mentioned "human emissions" started to decline in mid-2000s. How did they jointly lead to rapid increase in haze days?**

*Reply:*

It is true that human emissions have been reduced since mid-2000s, but **the pollution emissions still far exceed the capacity of atmospheric environment**, so it can **provide adequate particulate emissions** at all times to form haze pollution. This also contributes to the fact that haze pollution is extremely **sensitive** to meteorological conditions.

In February 2020, **the Ministry of Ecology and Environment** published an article focusing on the recent heavy air pollution in the Beijing-Tianjin-Hebei region and surrounding areas, with five experts explain the causes of the pollution **during COVID-19 quarantines** (http://www.mee.gov.cn/xxgk2018/xxgk/xxgk15/202002/t20200211_762584.html). Zifa Wang, a researcher at the Institute of Atmospheric Physics of the Chinese Academy of Sciences, explained that pollution emissions have fallen, but **not by nearly as much**

**as the environmental capacity**. Although social activities are at a low level, pollutant emissions are **still more than twice the environmental capacity**. Kebin He also said that under the current emissions intensity of the "2+26" region, although the total emissions are reduces, most of the pollutant emissions accumulated to the few city, **making the load of the atmospheric pollutants in several cities far exceeds the environmental capacity and causing heavy pollution**, such as Beijing, Tianjin and other cities. At the current level of emissions reduction, pollutants still exceed the capacity of the atmosphere, especially in adverse meteorological conditions. So anthropogenic emissions and climate factors jointly lead to rapid increase in haze days after 2010.

**Line 91: MLR should spell out.**

*Reply:*

We have added the full spelling of MLR to the text.

*Revisions:*

**Line 98:** In this study, the statistical model of fitted $HD_{NC}$ was built based on Multiple Linear Regression (MLR).

**Line 102: The trend should be given with its statistical significance. Please check throughout the text.**

*Reply:*

We have calculated a t test for trends, and given the statistical significance in the text.

*Revisions:*

**Line 110:** The trend of $HD_{NC}$ was vastly different before and after 2010: slowly decreased during 1991–2010 (P1) with a rate of 4.67 days/10 yr but rapidly increased after 2010 (P2, 2010–2018) with a rate of 25.43 days/10 yr, both of them passing 95% t test.

**Line 177-179:** Thus, the persistent decline in $SST_P$ during P1 (at a significant rate of $-0.2$ °C/10 yr, passing 95% t test; Table 1) contributed to the slowly decreasing trend of $HD_{NC}$ (Fig. 4a) via the modulations of $SST_P$ on the atmospheric circulation (Fig. S5).

During P2, the larger increase in $SST_P$ at a rate of 2.0 °C/10 yr (passing 95% t test) dramatically drove the rapid increase in $HD_{NC}$.

**Lines 193-194:** The $SST_A$ reached a infection point in 2010 (Fig. 4b) and contributed to the falling of $HD_{NC}$ during P1 (change rate of $SST_A$ = 0.55 °C/10 yr, passing 95% t test) and the rising of $HD_{NC}$ during P2 (change rate of $SST_A$ = –0.52 °C/10 yr, passing 95% t test).

**Lines 207-208:** The Snowc index fell slowly until 2010 (with a rate of –1.8%/10 yr, passing 95% t test) and then rose rapidly (with a rate of 28.3%/10 yr, , passing 95% t test) and experienced a large trend reversal in 2010……

**Lines 220-221:** The change rate of Soilw was 38.8 mm/10 yr passing 95% t test (opposite that of $HD_{NC}$) during P1, and the rate of change became more intense (–51.8 mm/10 yr, passing 95% t test) during P2……

**Lines 253-254:** More importantly, the fitted curve revealed a decreasing trend of HDNC (–5.24 days/10 yr, passing 95% t test)……

**Lines 109-110: Only PM2.5 data in Beijing used from the national measurement network? Or PM2.5 over North China? This should be also clarified in the caption of Figure 1.**

*Reply:*

The $PM_{2.5}$ concentration **over North China** were used from the national measurement network, we have clarified both in the text and the caption of Figure 1.

*Revisions:*

**Line 119:** ……and the $PM_{2.5}$ concentrations over North China monitored by China National Environmental Monitoring Centre from 2014 to 2018 ……

**Line 420-422:** Figure 1. ……The blue and green solid (dashed) lines indicate the number of days when the hourly $PM_{2.5}$ concentrations in a day exceeded 75 µg m$^{-3}$ and 100 µg m$^{-3}$, respectively, from 2009 to 2016 (2014 to 2018) using Beijing (North China) observed data from the US embassy (China National Environmental Monitoring Centre)…….

**Lines 136-137: I think they also show results with varying meteorology?**

*Reply:*

We have added the analysis of their results with varying meteorology in the text.

*Revision:*

**Lines 146-147:** The GEOS-Chem simulations with changing emissions and fixed meteorological conditions failed to reproduce the change trend of haze (Dang and Liao, 2019), but with varying meteorology and fixed emissions could recognize the interannual variation of haze days.

**Figure 1a: missing legend for green dash line. I don't think it is reasonable to show the red dash line. The sum of these pollutant emissions doesn't make sense.**
**Figure 1b: You should give observed trends of haze days along with the simulated trends.**

*Reply:*

In Figure 1a, we have added the legends for blue and green dash lines, and removed the red dash line representing the sum of these pollutant emissions. And we have added the observed trends of haze days in Figure 1b.

*Revision:*

[Figure]

**Figure 1.** (a) Variations in the December-January emissions (unit: Tg) of black carbon (BC), ammonia (NH$_3$), nitrogen oxide (NO$_x$), organic carbon (OC), sulfur dioxide (SO$_2$), PM$_{10}$ and PM$_{2.5}$ over North China from 1979 to 2013 and the variation in HD$_{NC}$ from 1979 to 2018 (black solid line). The blue and green solid (dashed) lines indicate the number of days when the hourly PM$_{2.5}$ concentrations in a day

exceeded 75 µg m$^{-3}$ and 100 µg m$^{-3}$, respectively, from 2009 to 2016 (2014 to 2018) using Beijing (North China) observed data from the US embassy (China National Environmental Monitoring Centre). (b) Temporal evolutions of HD$_{NC}$ (in black), simulated haze days (unit: days; red) in NC. The dashed lines denote linear regressions for 1991–2010 (P1) and 2010–2018 (P2). The black and red Trend 1 and Trend 2 represent the linear trends of the observed and simulated haze days in P1 and P2, respectively.